# Assembly of a Large Collection of Maxicircle Sequences and Their Usefulness for *Leishmania* Taxonomy and Strain Typing

**DOI:** 10.3390/genes13061070

**Published:** 2022-06-15

**Authors:** Jose Carlos Solana, Carmen Chicharro, Emilia García, Begoña Aguado, Javier Moreno, Jose M. Requena

**Affiliations:** 1Centro de Biología Molecular Severo Ochoa (CSIC-UAM), Departamento de Biología Molecular, Instituto Universitario de Biología Molecular (IUBM), Universidad Autónoma de Madrid, 28049 Madrid, Spain; jcsolana@cbm.csic.es; 2WHO Collaborating Centre for Leishmaniasis, Centro Nacional de Microbiología, Instituto de Salud Carlos III, Majadahonda, 28220 Madrid, Spain; cchichar@isciii.es (C.C.); egdiez@isciii.es (E.G.); 3Centro de Investigación Biomédica en Red de Enfermedades Infecciosas (CIBERINFEC), Instituto de Salud Carlos III, 28029 Madrid, Spain; 4Centro de Biología Molecular Severo Ochoa (CSIC-UAM), Genomic and NGS Facility (GENGS), 28049 Madrid, Spain; baguado@cbm.csic.es

**Keywords:** *Leishmania*, kDNA, maxicircle, phylogeny, trypanosomatids, mitochondrial DNA

## Abstract

Parasites of medical importance, such as *Leishmania* and *Trypanosoma,* are characterized by the presence of thousands of circular DNA molecules forming a structure known as kinetoplast, within the mitochondria. The maxicircles, which are equivalent to the mitochondrial genome in other eukaryotes, have been proposed as a promising phylogenetic marker. Using whole-DNA sequencing data, it is also possible to assemble maxicircle sequences as shown here and in previous works. In this study, based on data available in public databases and using a bioinformatics workflow previously reported by our group, we assembled the complete coding region of the maxicircles for 26 prototypical strains of trypanosomatid species. Phylogenetic analysis based on this dataset resulted in a robust tree showing an accurate taxonomy of kinetoplastids, which was also able to discern between closely related *Leishmania* species that are usually difficult to discriminate by classical methodologies. In addition, we provide a dataset of the maxicircle sequences of 60 *Leishmania infantum* field isolates from America, Western Europe, North Africa, and Eastern Europe. In agreement with previous studies, our data indicate that *L. infantum* parasites from Brazil are highly homogeneous and closely related to European strains, which were transferred there during the discovery of America. However, this study showed the existence of different *L. infantum* populations/clades within the Mediterranean region. A maxicircle signature for each clade has been established. Interestingly, two *L. infantum* clades were found coexisting in the same region of Spain, one similar to the American strains, represented by the Spanish JPCM5 reference strain, and the other, named “non-JPC like”, may be related to an important leishmaniasis outbreak that occurred in Madrid a few years ago. In conclusion, the maxicircle sequence emerges as a robust molecular marker for phylogenetic analysis and species typing within the kinetoplastids, which also has the potential to discriminate intraspecific variability.

## 1. Introduction

Leishmaniases are a group of parasitic diseases broadly distributed in most tropical and subtropical countries and the Mediterranean basin. More than 20 *Leishmania* species have been described as human pathogens, and the different clinical manifestations (cutaneous (CL), mucocutaneous (MCL) and visceral leishmaniasis (VL)) are mainly dependent on the species causing the infection [1]. Leishmaniasis is a major public health problem and the second most lethal parasitic disease, after malaria. In 2016, WHO reported that more than one billion people lived in *Leishmania* endemic regions and that every year 1 million CL, 300,000 VL, and 20–50,000 lethal VL cases occur [2] (and references therein).

The genus *Leishmania* is included in the order Trypanosomatida, together with other parasites of medical importance such as *Trypanosoma brucei* and *Trypanosoma cruzi*, which are the causative agents of sleeping sickness and Chagas disease, respectively [3,4]. A distinctive feature shared by trypanosomatids is the presence of the kinetoplast, a structure inside the mitochondrion containing a DNA network (known as kDNA) composed of a large number of circular DNA molecules. The kDNA network contains a small number of maxicircles, ranging from 20 to 40 kb (depending on the species), and thousands of minicircles, ranging from 0.5 to 2 kb [5]. Maxicircles are equivalent to the mitochondrial genomes present in most eukaryotes, and they encode for two mitochondrial rRNAs and several subunits of the electron transport chain complexes. Another peculiarity is that many maxicircle transcripts need to be edited by an RNA editing machinery that requires guiding of RNAs encoded in different minicircles [6].

Early phylogenetic studies of trypanosomatids were first based on the small subunit ribosomal rRNA (SSU rRNA) gene sequences, which allowed the identification of isolates at the level of genus and subgenus [7,8]. However, the SSU rRNA sequence evolution is very slow and the high sequence conservation does not allow very close species to be distinguished [9]. Nevertheless, the flanking genomic regions located between the SSU and 5.8S rRNAs, and between the latter and LSU rRNA, named internal transcribed spacer ITS1 and ITS2, respectively, are more variable in sequence and have been used to discriminate close *Leishmania* species and even subspecies [10,11,12]. Posterior typing approaches were based on a combination of several gene sequences, and strategies such as multilocus sequence typing (MLST) or microsatellite repeats (multilocus microsatellite typing, MLMT) are being widely used for the revision of *Leishmania* taxonomy [13,14,15] and parasite typing [16]. However, the use of phylogenies based on one or few molecular targets may be still inefficient in the discrimination of closely related organisms at the species level [17]. On the contrary, the application of phylogenetic analyses based on many molecular markers may be cumbersome if sequencing of those markers needs to be carried out individually for each organism [16].

The kDNA has been proposed as a discrete target for phylogenetic analyses [18,19] because this DNA is uniparentally inherited [20], and is not affected by polymorphisms and recombination events occurring in the *Leishmania* inter-species/-strain hybrids [21]. By comparison, whereas minicircles are heterogeneous in terms of both copy number and sequence, maxicircles represent reliable markers for phylogenetic analyses [19,22]. Maxicircles contain two different regions: the coding region (CR) and the divergent region (DR). The low sequence complexity and repeated nature of the DR have been insurmountable hurdles to determining its complete sequence and length until the use of long-range PCR [23], or a combination of short and long reads from massive sequence technologies [22,24]. However, the gene order (synteny) in the maxicircle CR is highly conserved among the trypanosomatids [19,23], a feature that enables maxicircles to be reliable markers to determine evolutionary relationships among members of this order of organisms [25]. Now, when whole-genome sequencing (WGS) technologies are widely used for determining the nuclear genomic sequences of an increasing number of *Leishmania* species and strains [26,27,28,29,30,31,32,33], and their raw data are in publicly available databases [34], it is feasible to use those data to assemble the corresponding maxicircle sequences.

In previous work, we described a pipeline designed to assemble the complete maxicircle sequence using next genome sequencing (NGS) reads, and the maxicircles for a few *Leishmania* species were determined [19]. Subsequently, the methodology has been used to determine the maxicircle sequences of several *L. infantum* strains [35] and the *T. cruzi* maxicircle sequence [22]. Here, we assembled a collection of maxicircle sequences using publicly available WGS data, and generated a dataset consisting of 26 sequences that may be useful to conduct phylogenetic studies in trypanosomatids. Moreover, we showed that maxicircle CR sequences also serve to discriminate among closely related species of the same *Leishmania* complex. In addition, we confirmed that even this molecular target may be useful for grouping different *L. infantum* clinical isolates according to their geographical origin and to establish phylogenic relationships.

## 2. Materials and Methods

### 2.1. Genomic Sequences and Data Availability

The origin of the sequencing WGS reads used for the 86 maxicircle assemblies reported in this work and other previously available sequences are described in Table 1 and Appendix A. The assembled sequences were deposited in the Mendeley Data repository (https://data.mendeley.com/, accessed on 12 June 2022) with the accession numbers indicated in Table 1 and in Appendix A.

### 2.2. Assembly of Maxicircle Sequences

The bioinformatic pipeline and the associated in-house scripts have been described elsewhere [19]. Briefly, WGS reads were aligned against the reference nuclear genome using Bowtie2 in order to separate the non-aligned reads. The *Leishmania* genome files were retrieved from either the TriTrypDB database [34] or the Leish-ESP web page (http://leish-esp.cbm.uam.es, accessed on 8 May 2022). After a quality filtering using Prinseq, a de novo assembly was performed with IDBA-UD, and contigs longer than 500 nucleotides were analyzed using the NCBI-BLAST tool to evaluate sequence identities with reported maxicircle sequences. Only contigs that included the complete CR of maxicircles were used for phylogenetic analyses. The CR sequences of the maxicircles were established according to the positions of the genes 12S rRNA (beginning) and ND5 (end), using the *L. tarentolae* maxicircle annotations as reference (GenBank: M10126.1).

### 2.3. Phylogenetic Analysis

Phylogenetic trees were constructed based exclusively on the CR sequence of the maxicircles. Firstly, the sequences were aligned using the Clustal W algorithm and the following parameters: Gap Opening Penalty 15, Gap Extension Penalty 6.66, DNA Weight Matrix ClustalW 1.6, Transition Weight 0.5, Delay Divergent Cutoff 30%. Phylogenetic relationships were inferred using the maximum likelihood method and Tamura-Nei model [43]. Initial trees for the heuristic search were obtained automatically by applying neighbor-joining (NJ) and BioNJ algorithms to a matrix of pairwise distances estimated using the Tamura–Nei model, and the topology was then selected using the superior log likelihood value. The bootstrap consensus trees that represent the evolutionary history of the analyzed taxa were inferred from 1000 replicates [44]. All the evolutionary analyses were conducted in MEGA 11 [45].

## 3. Results and Discussion

### 3.1. Creation of a Maxicircle Dataset for Trypanosomatids

The structure of the *L. tarentolae* mitochondrial genome (maxicircle and minicircles) is one of the best characterized to date among the trypanosomatids [46,47]. A significant portion of the maxicircle (around 12 kb), named “divergent region” or DR, consists of low-complexity, tandemly repeats and its complete sequence remains undetermined for most species, but it is considered to be non-coding [48]. In contrast, the “coding region” (CR), completely sequenced years ago [49], has a remarkable conservation, in terms of both sequence and gene order among the trypanosomatids [19,22,25]. Figure 1 illustrates the absolute conservation in gene order (and strain polarity) in the maxicircles of evolutionary distant species such as *L. tarentolae*, *L. infantum*, and *T. cruzi.* Taking into account this remarkable conservation in the gene organization among the mitochondrial genomes (maxicircles) of trypanosomatids, we and others proposed that the complete CR of maxicircles represents an excellent phylogenetic marker for conducting evolutionary analyses among species within this order of protists [19,23]. The maxicircle CR may be considered to be a “natural catenation” of genes equivalent to the “artificial catenation” followed in the MLST strategy. Moreover, the fact that most maxicircle-encoded transcripts require specific addition and deletion of uridine residues by the RNA editing machinery to form translatable mRNAs introduces an additional evolutionary variability among the orthologous genes. Notably, a maxicircle gene, cytochrome b (Cyb; Figure 1), is a classic and widely used marker for phylogenetic studies [50,51], but it is known that cytochrome b sequences cannot discriminate species such as *L. donovani* and *L. infantum*, or *L. braziliensis* and *L. peruviana* [10].

Until recently, only a small number of maxicircle sequences was available, most of which had an incomplete CR. The main reasons for this were the relatively long size (around 16 kb) of the CR and the difficulty in obtaining pure preparations of mitochondrial DNA. Nevertheless, advances in sequencing methodologies have contributed to solving these hurdles and, as we recently demonstrated [19], it is possible to assemble the complete CR of the maxicircles by exploiting the sequence reads derived from whole-DNA sequencing (WGS) projects. As there are many publicly available trypanosomatid WGS data (and this quantity is exponentially increasing), we considered the possibility of creating datasets of maxicircle sequences for those trypanosomatid species. These datasets may be useful for conducting future phylogenetic studies in which the aim is to define the taxonomic location of new species and/or strain typing.

Table 1 lists a compilation of 26 maxicircles for which a complete CR sequence could be determined. Eighteen were previously described in articles and/or deposited in databases. Additionally, for this work, we assembled another eight sequences using raw sequencing reads deposited in public repositories (the accession numbers are included in Table 1). Moreover, we created individual Mendeley data for every sequence in order to make them easily accessible (the links to the corresponding Mendeley entries are also included in Table 1). In this manner, this dataset represents a ready-to-use collection of maxicircle sequences covering a wide spectrum of species within the order of trypanosomatids.

### 3.2. Validation of In Silico Assembled Maxicircle CR for Species Identification and Phylogenetic Analyses

To illustrate the robustness of maxicircle sequences for taxonomic purposes, the entire CR sequence of the maxicircles listed in Table 1 were aligned and used to construct a phylogenetic tree by the maximum likelihood (ML) method (Figure 2). The results showed a total agreement with the phylogenetic tree generated from either a concatenated dataset of 42 protein sequences from an equivalent group of trypanosomatids [52], a study based on alignments of concatenated *Leishmania* gene sequences (i.e., 18S rRNA, gGAPDHm RPOIILS, and HSP70) [53] or the alignment-free analysis of frequency feature profiles (FFP) based on a wide range of trypanosomatid maxicircle sequences [25]. Moreover, the bootstrap analysis indicated an extremely robust structure of the tree branches, which are supported by 99–100% bootstrap confidence values for the separation of all the trypanosomatid species included in the study.

As expected, the species of genera *Leishmania* and *Trypanosoma* formed two separate clades. Within the genus *Trypanosoma*, *T. cruzi* appeared close to *T. rangeli*, a human non-pathogenic species, in agreement with previous phylogenetic analysis based on HSP70 gene [54]. Moreover, other *Trypanosoma* species close to *T. cruzi* are *Trypanosoma copemani* and *Trypanosoma lewisi* [40]. Finally, the African trypanosomes *Trypanosoma vivax* and *T. brucei* form two paraphyletic groups, in which *T. brucei* is more closely related to *T. cruzi* than *T. vivax*, in agreement with previous studies [25]. 

In the phylogenetic tree, the monoxenus species *A. daenei* (formerly *Crithidia deanei*), a representative member of the subfamily Strigomonanidae, appeared as an isolated branch between both clades, although it is closest to the *Leishmania* clade. 

Within the Leishmaniiae subfamiliy, the monoxenus parasite *Zelonia* appeared separately from the branch formed by the different *Leishmania* species analyzed, according to previous studies that located the genus *Zelonia* in a subclade distinct to that formed by the dixenous parasites of the genera *Leishmania*, *Endotrypanum*, and *Porcisia* [53]. Within the *Leishmania* genus, the species appeared grouped in agreement with the four established subgenera, i.e., *Leishmania, Viannia, Sauroleishmania*, and *Mundinia* [16]. The phylogenetic analysis derived from the assembled maxicircle sequences was able to divide the species of the subgenus *Viannia*, restricted to the Neotropics (New World), into two complexes that include the very close related *L. braziliensis* and *L. peruviana* species (the *L. braziliensis* complex) [55], and the *L. guyanensis* complex, which includes also *L. panamensis* [16]. The species *L. lainsoni*, belonging also to the *Viannia* subgenus, appeared in the same phylogenetic branch. Similarly, within the subgenus *Leishmania*, the phylogenetic tree also showed a clear separation among the different complexes: *L. major*, *L. tropica*, *L. donovani*, and *L. mexicana* [56]. The *L. donovani* complex includes the two species traditionally associated with visceral leishmaniasis, *L. donovani* and *L. infantum*. The maxicircle sequences were able to distinguish between different species of the *L. tropica* complex (*L. tropica* and *L. aethiopica*). The representative of the *L. mexicana* complex (*L. amazonensis*) appeared in a separate branch. Similarly, a distinct branch is formed by the species *L. adleri* and *L. tarentolae*, which are grouped within the *Sauroleishmania* subgenus. Finally, in the tree, *L. enrietti* and *L. martiniquensis* formed a separate branch; these species belong to the subgenus *Mundinia*, which includes parasites exhibiting a high phenotypic plasticity and being able to infect a wide range of hosts, including humans [57].

### 3.3. Validation of In Silico Assembled Maxicircle CR for Species Identification and Phylogenetic Analyses

The species *L. infantum* and *L. donovani* are hardly distinguished by phylogenetic analyses based on single molecular markers [58]. The separation of both species and the identification of different populations were only attained after the application of whole genomic approaches [59,60]. Remarkably, in the phylogenetic tree based on the maxicircle CR sequences (Figure 2), both species appeared in separate branches supported by a bootstrap value of 100. This observation prompted us to analyze whether the maxicircle sequence would also be useful for determining intraspecific variability. To address this question, maxicircle sequences from a collection of *L. infantum* field isolates from distinct geographical origins or phenotypes were gathered. 

For the study, WGS genomic reads (DNA-seq) for 60 *L. infantum* field isolates, generated by several research groups and deposited in public databases, were used. Thus, using Illumina reads deposited in the ENA project PRJNA589999 [61], 28 complete maxicircle CR sequences were assembled and compiled in Mendeley Data [62]. Another three maxicircles were assembled from the NGS data generated by Bussotti et al. [63] and compiled in Mendeley Data [64]. Similarly, based on the DNA-seq data generated in the works of Rogers et al. (PRJEB2473; [21]) and Franssen et al. (PRJEB2600; [59]), we assembled three [65] and four [66] maxicircle-complete CR sequences, respectively. Moreover, we compiled in a Mendeley dataset [67] the seven *L. infantum* maxicircle sequences reported by Bussotti et al. (PRJNA607007; [35]). Finally, using DNA-seq data from our on-going study of the genomic assembly of a collection of *L. infantum* Spanish isolates, we assembled complete CRs for another 14 maxicircles [68].

A phylogenetic tree including all the *L. infantum* strains, in which the complete maxicircle CR sequence could be assembled, is shown in Figure 3 (panel A). Remarkably, all the *L. infantum* isolates were distantly grouped from the two *L. donovani* maxicircle sequences included in the analysis; this finding, in addition to supporting the correctness of the species typing, validates the use of the maxicircle as a marker for distinguishing the two species forming the *L. donovani* complex, i.e., *L. donovani* (sensu stricto) and *L. infantum*.

Regarding the distribution of the *L. infantum* strains in the tree, all the isolated obtained from Brazilian patients (isolates P3170, P3167, etc.) [61,63] formed a monophyletic sub-clade. The closest relatives to these Brazilian isolates are a group of strains found in Europe, including the *L. infantum* JPCM5 reference strain [31], which was isolated in Spain (Figure 3). This result is in agreement with previous data that point to the exportation of this species from South-Western Europe to the New World 500 years ago [69]. Thus, we dubbed these European *L. infantum* isolates the “JPC-like” group.

Significantly distant from the “JPC-like” group, three additional *L. infantum* clades appeared as separated branches in the tree and were supported by high bootstrap values (96–100) (Figure 3). The isolates CUK1, CUK4, and CUK12 were obtained from patients in the Çukurova province of Turkey, where cutaneous leishmaniasis (CL) rather than visceral leishmaniasis (VL) outcomes were observed [70]. It has been suggested that these strains may have originated from a hybridization event between two different species of the *L. donovani* complex, one of which is closely related to the JPCM5 strain [21]. Nevertheless, in posterior analyses in which a large number of strains of the *L. donovani* complex was studied, the authors suggested that the origin of this group may be a hybridization between a Chinese *L. infantum* strain and an isolate from Cyprus [59]. In this regard, it is noteworthy that two isolates (CH32 and CH34) that were grouped close to the Turkish CUK strains were also isolated from CL cases in Cyprus [71]. In contrast, two other Cypriot isolates analyzed in this work, CH33 and CH36, which were derived from VL patients, form a different clade/branch in the tree; nevertheless, the third isolate forming this clade (CH35; Figure 3) was reported to be derived from a patient with CL. Interestingly, the grouping of the isolates CH33, CH35, and CH36 was also noted in a study based on MLMT analyses [71] and in a genome-wide SNP-based phylogenetic analysis [59]. In those studies, these isolates were suggested to represent the parent genotype of CUK strains.

Finally, some of the Spanish isolates (LLM-2409, LLM-2406, LLM-2408, LLM-2404, and LLM-2410) were found to form a distinct clade, clearly separated from the JPC-like group (Figure 3). LLM-2409 and LLM-2404 were isolated from CL cases and the rest of isolates were obtained from VL samples (LLM-2406 and LLM-2410 were isolated from HIV-immunocompromised individuals). Our analysis points to the existence of a clear phylogenetically separated group overlapping geographically with the JPC-like group. Some years ago, a molecular typing by MLMT was performed in parasites isolated from the same region during the largest reported community outbreak of human leishmaniosis in Europe [72]. Although seven different genotypes were identified in that study, only one genotype was responsible from most of the outbreak cases. Nevertheless, none of these genotypes was considered to be new or different from those parasites isolated years before the outbreak. According to MLMT data, those genotypes were grouped into two categories: ITS-A type (associated with MON-1 group) and the ITS-Lombardi type, which was present in all MON-24 strains and 20% of MON-1 isolates. These Lombardi-type isolates, which accounted for 71% of the parasite lines obtained from the outbreak cases, clearly clustered in a different clade than the ITS-A-type samples [72]. Interestingly, the conclusions derived from the phylogenetic features of the isolates included in the present work, obtained several years after the outbreak, agree with previous observations. Thus, the LLM strains that belong to the “JPC-like” group also belong to the ITS-A type, whereas the LLM isolates of the “non-JPC” group belong to the Lombardi-type (except LLM-2404, which is an ITS-A type) (Figure 3). Together, these results suggest the existence of an evolutionary leap occurring in this region, leading to the emergence of the “non-JPC” group, presumably by a hybridization event between a JPC-like strain and an unidentified *L. infantum* parent strain. This possibility is also supported by the fact that all the non-JPC LLM strains showed a larger number of polymorphic sites (unpublished laboratory data), whereas the JPC-like LLM strains maintained the low level of heterozygosity that has been described previously for this species [59]. The Lombardi genotype has been circulating in the region since at least 1987 [72], and its infectivity and transmission capabilities have been studied [73,74]. Hybridization between *L. infantum* strains has been reported in Tunisia [75], and between *L. infantum* and *L. major* in Portugal [76]. In summary, this is the first time that, using maxicircle sequences, solid genetic evidence of two *L. infantum* groups co-existing in endemic regions of Spain has been documented. Nevertheless, the nature of these isolates and possible hybrid origin needs to be addressed by comparative analyses involving not only kDNA sequences, but also the nuclear DNA sequences, following the approach reported in a recent study [77].

To delve deeper into the outsider Spanish non-JPC LLM isolates group (LLM-2409, LLM-2406, LLM-2408, LLM-2410) we closely examined the alignment of the maxicircle sequences. We found 17 polymorphic positions in the CR sequence among the Spanish *L. infantum* isolates. Of note, 11 of these positions are common among the isolates of the non-JPC group and may be considered to be a sequence signature for the group. These polymorphic nucleotides are located in the genes 12sRNA, CYb, ATPase 6, ND2, ND1, CO2, MURF2, and CO1 (Appendix A. This signature may be helpful as a marker to easily identify field isolates that belong to this outsider group using conventional molecular tools such as PCR. 

Additionally, maxicircle signatures could be obtained for other *L. infantum* clades such as those defined by the Turkish and the Cypriot isolates (Appendix A). Interestingly, a relationship between Eastern European isolates and the non-JPC LLM group may exist because four positions of the non-JPC signature are found in the CH33, CH35, and CH36 clade (postulated parent strains for other CH and CUK hybrid isolates, as discussed above). These polymorphic positions that are common in the Western Mediterranean and Eastern Mediterranean isolates were found in genes ND2, ND1, CO2, and CO1 (Appendix A). Thus, these genes may be considered to be potential targets to find possible non-JPC *L. infantum* strains. Nevertheless, WGS analysis of the nuclear genome should be used to thoroughly study if a hybrid origin of the non-JPC group can be considered, or if the genetic differences are caused by natural divergence.

## 4. Conclusions

The maxicircle sequence contains enough information not only to distinguish between closely related species, but also to discriminate different groups of strains within a given species. The phylogenetic results are superior to those derived by ITS1 MLST or by the use of several gene markers, and close to those including whole-genome analysis [18]. However, the analysis of whole nuclear genomes demands advanced computational resources and the use of diverse bioinformatics tools to handle such a huge amount of data, and, in consequence, this approach has serious limitations for most clinical laboratories.

Before developing the current whole-genome sequencing (WGS) technologies, the determination of the complete maxicircle CR sequence was a cumbersome task. However, as shown here, the assembly of this molecule from the WGS data is easy. Hence, considering that maxicircles may become a highly used phylogenetic marker for taxonomy of trypanosomatids, we created a collection of maxicircle sequences covering a wide spectrum of organisms within this order. Moreover, we showed that maxicircle-based phylogenetic analyses also serve to uncover evolutionarily distinct groups within a given species, as exemplified here for *L. infantum* strains. In this regard, we determined the existence of a close relationship between South American and Mediterranean isolates, but, at the same time, our data support the existence of two groups of strains coexisting in Spain.

Full-genome sequencing is becoming a routine technique that is provided by both public institutes and private companies, with costs affordable for most research and diagnostic laboratories. Moreover, there are several public repositories (ENA, NCIB, and EGA) in which raw sequencing reads may be stored. Our recommendation is that researchers should be aware that these data, without further analysis, are extremely valuable and useful for current and future studies. Therefore, we encourage researchers to deposit this information, which, by itself, would represent a significant contribution to the scientific community working on leishmaniasis.

## Figures and Tables

**Figure 1 genes-13-01070-f001:**
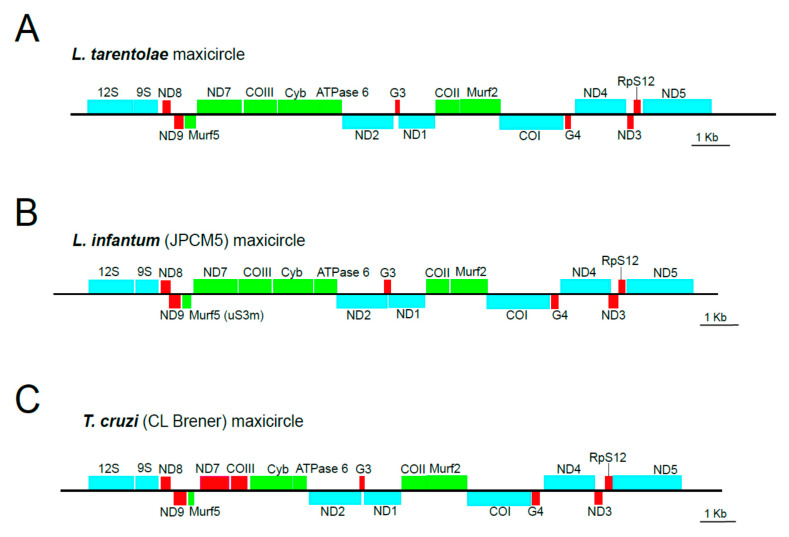
Schematic diagram of the coding region of *L. tarentolae* (**A**), *L. infantum* (**B**), and *T. cruzi* (**C**) maxicircles. The nomenclature and abbreviations are those established for *L. tarentolae.* Blue, green, and red boxes are used to indicate non-edited genes, edited genes, and pan-edited genes, respectively. The drawings are based on the gene coordinates reported elsewhere: *L. tarentolae* (GenBank ID M10126 and [47], *L. infantum* [19], and *T. cruzi* [42].

**Figure 2 genes-13-01070-f002:**
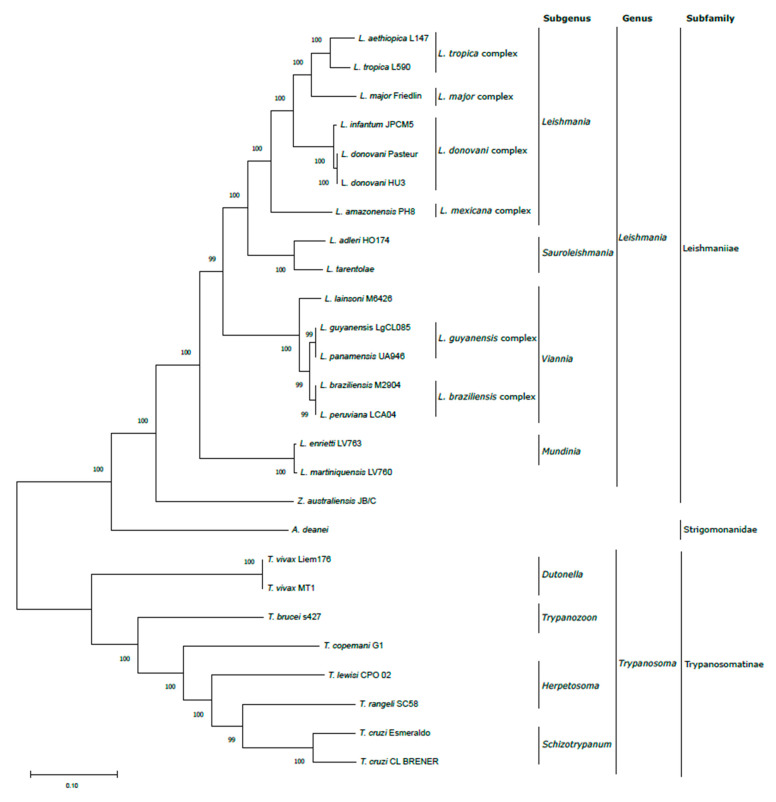
Evolutionary analysis of the Trypanosomatidae family based on the CR of the maxicircle sequence. The evolutionary history was inferred using the maximum likelihood method and the Tamura–Nei model (Tamura and Nei 1993). The tree with the highest log likelihood (−166,927.58) is shown. The percentage of trees in which the associated taxa are clustered together is shown next to the branches. The tree is drawn to scale, with branch lengths representing the number of substitutions per site. This analysis involved 26 nucleotide sequences. Codon positions included were 1st + 2nd + 3rd + Noncoding. There was a total of 16,895 positions in the final dataset. All the evolutionary analyses were conducted in MEGA 11 [45].

**Figure 3 genes-13-01070-f003:**
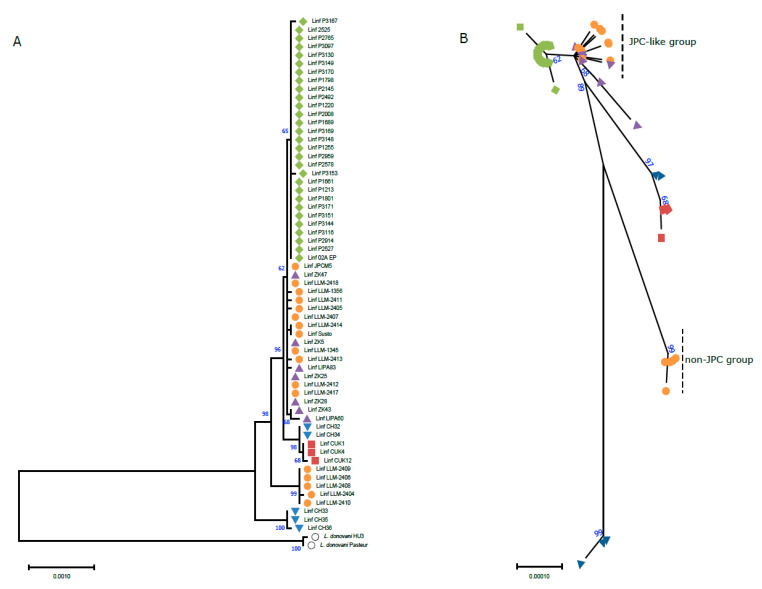
Phylogenic relationship among Mediterranean and American *L. infantum* isolates (**A**,**B**). The evolutionary history was inferred using the maximum likelihood method and the Tamura–Nei model [43]. The tree with the highest log likelihood (−21,148.45) is shown. The tree is drawn to scale, with branch lengths representing the number of substitutions per site. This analysis involved 63 nucleotide sequences. Codon positions included were 1st + 2nd + 3rd + Noncoding. There was a total of 16,285 positions in the final dataset. 
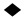
 Brazilian isolates; 
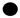
 Spanish isolates; 
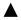
 Tunisian isolates; 
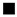
 Turkish isolates; 
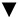
 Cypriot isolates; 
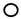

*L. donovani* (outgroup). All the evolutionary analyses were conducted in MEGA 11 [45].

**Table 1 genes-13-01070-t001:** Maxicircle sequence datasets from trypanosomatid species used for phylogenetic studies.

Species	GenBank/ENA *	Reference	Mendeley Data (DOI) ^a^
*Leishmania aethiopica* L147 MHOM/ET/1972/L100	SRR834802	This work	10.17632/8h8tzrbzft.1
*Leishmania tropica* L590 MHOM/IL/1990/P283	SRR834907	This work	10.17632/23t2h58cvr.1
*Leishmania major* MHOM/IL/81/Friedlin	LR697138	[19]	10.17632/hdyj8hbt39.1
*L. infantum* JPCM5 MCAN/ES/98/LLM-724	LR697137	[19]	10.17632/nszm7rb8y7.1
*Leishmania donovani* Pasteur	CP022652.1	Direct submission	10.17632/jhy362m5ms.1
*Leishmania donovani* HU3 MHOM/ET/67/HU3	ERR2191875	This work	10.17632/74d3b3pnvt.1
*Leishmania amazonensis* IFLA/BR/1967/PH8	SRR8584809	This work	10.17632/pfdfzpwgjd.1
*Leishmania adleri* MARV/ET/75/HO174	LR697136	[19]	10.17632/98cn9h5t67.1
*Leishmania tarentolae*	M10126.1	[36]	10.17632/6kwj82nt8s.1
*Leishmania guyanensis* LgCL085	LR697135	[19]	10.17632/rcjzz74fvj.1
*Leishmania panamensis* UA946	MK570510.1	[37]	10.17632/9sxnsjy9xx.1
*Leishmania braziliensis* MHOM/BR/75/M2904	LR697134	[19]	10.17632/bg3 × 4tcr64.1
*Leishmania peruviana* MHOM/PE/90/LCA0482	ERR3656053	This work	10.17632/4wf4hb3k7g.1
*Leishmania lainsoni* MHOM/BR/1981/M6426	SRR1657912	This work	10.17632/k6ffm6j8c3.1
*Leishmania enrietti* LV763 MCAV/BR/2001/CUR178	SRR13558795	This work	10.17632/xhj67pv882.1
*Leishmania martiniquensis* LV760 MHOM/TH/2012/LSCM1	SRR13558784	This work	10.17632/9dykktyxp5.1
*Zelonia australiensis* JB/C	MK514117.1	[23]	10.17632/c4zvyzftfg.1
*Angomonas deanei*	KJ778684.1	Direct submission	10.17632/88w6bk3mkr.1
*Trypanosoma vivax* Liem176	KM386509.1	[38]	10.17632/484wfgchdr.1
*Trypanosoma vivax* MT1	KM386508.1	[38]	10.17632/wmkxhtz5rg.1
*T. brucei brucei* s427	M94286.1	[39]	10.17632/7yrtcn4nkk.1
*Trypanosoma copemani* G1	MG948557.1	[40]	10.17632/g44sr7djp4.1
*Trypanosoma lewisi* CPO 02	KR072974.1	[41]	10.17632/6w4gdb9rg8.1
*Trypanosoma rangeli* SC58	KJ803830.1	Direct submission	10.17632/gs6sbtbh7z.1
*T. cruzi* Esmeraldo	DQ343646.1	[42]	10.17632/7hfpy5frmv.1
*T. cruzi* CL Brener	DQ343645.1	[42]	10.17632/7gtbgwjjv8.1

* The NCBI GenBank accession numbers are included for published maxicircle sequences. For new assembled maxicircles (this work), the ENA accession numbers corresponding to whole-genome reads are provided. ^a^ Accessed on 12 June 2022.

## Data Availability

The assembled sequences were deposited in the Mendeley Data repository (https://data.mendeley.com/, accessed on 12 June 2022) with the accession numbers indicated in Table 1 and in the Appendix A.

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
