# Peer review of "Assembly of a Large Collection of Maxicircle Sequences and Their Usefulness for Leishmania Taxonomy and Strain Typing"

_genes, 2022, doi:10.3390/genes13061070_

Round 1

Reviewer 1 Report

In this study, authors assembled the complete coding region of the maxicircles for 26 species of trypanosomatids. Phylogenetic data showed an accurate taxonomy for this group of organisms. According to the data presented, those sequences may discriminate closely related Leishmania species that are usually difficult to discriminate by classical methodologies. The results presented in the manuscript are clear and very well presented; however, some minor points should be considered.

1) In the introduction, authors state SSU rRNA sequences are highly conserved and do not allow distinguishing very close related species (lines 61-63). However, ITS sequences may be useful for discrimination close related species, including species of the subgenus Viannia. Please check Espada et al., 2021 [Acta Tropica]; Van der Auwera et al., 2014 [Journal of Clinical Microbiology]; Van der Auwera, Dujardin, 2015 [Clinical Microbiology Reviews]. I would suggest to include this information in the introduction of the manuscript.

2) Similarly, hsp70 sequences have been also demonstrated to discriminate Leishmania species, but not at intraspecies level. Based on that, it would be interesting if authors included in the manuscript which potential genes of maxicircles coding regions could be useful for species discrimination? Is there any one specifically? Or only the full maxicircle coding region sequences are useful for species discrimination? This data would be particularly interesting for those that work with Leishmania species typing.

3) Still based on that was mentioned before, it is known that cytochrome b sequences cannot discriminate species like L. donovani and L. infantum, nor L. braziliensis and L. peruviana. Please check Van der Auwera, Dujardin, 2015 [Clinical Microbiology Reviews] and statement at page 4, lines 149-150. I suggest to review this statement.

Author Response

In this study, authors assembled the complete coding region of the maxicircles for 26 species of trypanosomatids. Phylogenetic data showed an accurate taxonomy for this group of organisms. According to the data presented, those sequences may discriminate closely related Leishmania species that are usually difficult to discriminate by classical methodologies. The results presented in the manuscript are clear and very well presented; however, some minor points should be considered.

We appreciate this general comment about the value of our work, and we are grateful for the helpful comments and suggestions.

1) In the introduction, authors state SSU rRNA sequences are highly conserved and do not allow distinguishing very close related species (lines 61-63). However, ITS sequences may be useful for discrimination close related species, including species of the subgenus Viannia. Please check Espada et al., 2021 [Acta Tropica]; Van der Auwera et al., 2014 [Journal of Clinical Microbiology]; Van der Auwera, Dujardin, 2015 [Clinical Microbiology Reviews]. I would suggest to include this information in the introduction of the manuscript.

In agreement with the reviewer’s suggestion, we have added this relevant information to the revised manuscript.

2) Similarly, hsp70 sequences have been also demonstrated to discriminate Leishmania species, but not at intraspecies level. Based on that, it would be interesting if authors included in the manuscript which potential genes of maxicircles coding regions could be useful for species discrimination? Is there any one specifically? Or only the full maxicircle coding region sequences are useful for species discrimination? This data would be particularly interesting for those that work with Leishmania species typing.

We have not analysed the contribution of each one of the 19 genes comprising the maxicircle coding region (CR) in the belief that "The whole is more than the sum of the parts". According to many articles in the field, it is likely that a sole gene would not allow discrimination between very close species (this is in agreement with the reviewer’s comment in point 3, see below). The purpose of our work was not to find a maxicircle gene for Leishmania typing purposes, our suggestion is to use the complete CR sequence (in the context that a large number of NGS data are being generated for Leishmania isolates). For those researchers looking for a single molecular marker (that may be directly PCR amplified and sequenced from a given isolate), our suggestion is to use one among the many well-characterized molecular markers described in the literature and listed in comprehensive reviews (Van der Auwera, Dujardin, 2015, Clinical Microbiology Reviews; Kuhls, Mauricio, 2019, Methods Mol Biol]).

3) Still based on that was mentioned before, it is known that cytochrome b sequences cannot discriminate species like L. donovani and L. infantum, nor L. braziliensis and L. peruviana. Please check Van der Auwera, Dujardin, 2015 [Clinical Microbiology Reviews] and statement at page 4, lines 149-150. I suggest to review this statement.

We have included this comment in the revised manuscript. Thanks

Reviewer 2 Report

Dear Editor;
Some minor grammatical errors in the article should be corrected. I have already fixed some of these errors. Also, it is useful to have your work checked by a native speaker.
In the introduction part, information such as the relationship between the Leishmaniasis group and public health, disease factors and percentage should be included.
In the Material and Methods section; Information and technical features of all products used in the sequencing process, such as barcode, brand, web access, package program, if any, should be given.
The resolution of Figure 3 should be increased if possible.
In the conclusion section, the messages to be given to the reader should be included, not the conclusions.

Line 20 promissing should be changed as promising
Line 43 in the Mediterranean…  (in should be deleted)
Line 82 whole genome should be changed as  whole-genome
Line 87 In a previous…. (a should be deleted)
Line 88 as a result  (, should be deleted)
Line 102  previous should be changed as  previously
Line 184  were should be changed as  was
Line 204 subfamiliy should be changed as  subfamily
Line 211 in should be changed as  into

Author Response

We are grateful for the detailed revision of the text made by the reviewer. Apart from those errors marked by the reviewer (see below), we have read carefully the revised version of the manuscript to avoid as many as possible grammatical errors. Fortunately, and this is a benefit of publishing in Genes, the Editorial team also revise the manuscript looking for grammatical issues before sending the manuscript to print.

In the introduction part, information such as the relationship between the Leishmaniasis group and public health, disease factors and percentage should be included.

We have included a short paragraph including relevant data that may allow to the reader be aware of the important public health representing leishmaniasis. Thanks for the suggestion.

In the Material and Methods section; Information and technical features of all products used in the sequencing process, such as barcode, brand, web access, package program, if any, should be given.

For this work, we have not produced any new sequence data. We have used WGS data available in public repositories. The accession codes for those WGS data have been included in the text and tables. In this work, based on the available information, we have assembled the maxicircle sequences for a large number of Leishmania species and strains. The methodological pipeline is described in section 2.2.

The resolution of Figure 3 should be increased if possible.

In agreement with the reviewer’s comment, the resolution of figure 3 has been improved.

In the conclusion section, the messages to be given to the reader should be included, not the conclusions.

Following the reviewer’s suggestion, a final recommendation has been added to the Conclusions section.

Line 20 promissing should be changed as promising

Done

Line 43 in the Mediterranean…  (in should be deleted)

Done

Line 82 whole genome should be changed as  whole-genome

Done here and in other two places.

Line 87 In a previous…. (a should be deleted)

Done

Line 88 as a result  (, should be deleted)

Done

Line 102  previous should be changed as  previously

Done

Line 184  were should be changed as  was

Done. Many thanks for this careful revision of the text.